# The Impact of Salinity on Crop Yields and the Confrontational Behavior of Transcriptional Regulators, Nanoparticles, and Antioxidant Defensive Mechanisms under Stressful Conditions: A Review

**DOI:** 10.3390/ijms25052654

**Published:** 2024-02-24

**Authors:** Mostafa Ahmed, Zoltán Tóth, Kincső Decsi

**Affiliations:** 1Festetics Doctoral School, Institute of Agronomy, Georgikon Campus, Hungarian University of Agriculture and Life Sciences, 8360 Keszthely, Hungary; 2Department of Agricultural Biochemistry, Faculty of Agriculture, Cairo University, Giza 12613, Egypt; 3Institute of Agronomy, Georgikon Campus, Hungarian University of Agriculture and Life Sciences, 8360 Keszthely, Hungary; toth.zoltan@uni-mate.hu (Z.T.); szaszkone.decsi.eva.kincso@uni-mate.hu (K.D.)

**Keywords:** salinity, crops, reactive oxygen species (ROS), enzymatic and non-enzymatic anti-oxidative system, abiotic stress, osmotic stress, transcriptional factors, nanoparticles, gene expression

## Abstract

One of the most significant environmental challenges to crop growth and yield worldwide is soil salinization. Salinity lowers soil solution water potential, causes ionic disequilibrium and specific ion effects, and increases reactive oxygen species (ROS) buildup, causing several physiological and biochemical issues in plants. Plants have developed biological and molecular methods to combat salt stress. Salt-signaling mechanisms regulated by phytohormones may provide additional defense in salty conditions. That discovery helped identify the molecular pathways that underlie zinc-oxide nanoparticle (ZnO-NP)-based salt tolerance in certain plants. It emphasized the need to study processes like transcriptional regulation that govern plants’ many physiological responses to such harsh conditions. ZnO-NPs have shown the capability to reduce salinity stress by working with transcription factors (TFs) like AP2/EREBP, WRKYs, NACs, and bZIPs that are released or triggered to stimulate plant cell osmotic pressure-regulating hormones and chemicals. In addition, ZnO-NPs have been shown to reduce the expression of stress markers such as malondialdehyde (MDA) and hydrogen peroxide (H_2_O_2_) while also affecting transcriptional factors. Those systems helped maintain protein integrity, selective permeability, photosynthesis, and other physiological processes in salt-stressed plants. This review examined how salt stress affects crop yield and suggested that ZnO-NPs could reduce plant salinity stress instead of osmolytes and plant hormones.

## 1. Introduction

A global scarcity of water resources, pollution, and an increase in soil and water salinization mark the beginning of the twenty-first century. A rise in human population and a decline in the land available for cultivation threaten agricultural sustainability [1]. High winds, harsh temperatures, soil salinity, drought, and flooding are only some of the environmental challenges affecting agricultural crop productivity and cultivation [2]. Regarding the quality and quantity of crops produced, soil salinity is among the worst ecological pressures [1]. A saline soil is defined as having an electrical conductivity (EC) of the saturation extract (ECe) at the root zone of more than 4 dS m^−1^ (about 40 mM NaCl) at 25 °C and an exchangeable salt content of 15% [3]. Most crop plants have a decreased yield at this ECe, even though many crops exhibit yield losses at lower ECes [2]. Based on estimations, high salinity has impacted around 33% of irrigated agricultural fields and 20% of the total farmed land globally [4]. Furthermore, the expansion of salinized lands at a rate of 10% per year [5] can be attributed to various factors, including insufficient precipitation, excessive surface evaporation, irrigation with salinized water, and suboptimal cultural practices. According to projections, the global proportion of arable land affected by salinization is anticipated to surpass 50% by 2050 [6].

Crop plants’ physiological and biochemical pathways are adversely affected by soil salinity through a complex mechanism [7]. When there is too much Na^+^ in the cell, cytosolic K^+^ and Ca^2+^ leave the cell. This upsets the balance of their homeostasis, which leads to nutritional deficiencies, oxidative stress, slowed growth, and cell death [8]. Several stomatal restrictions, such as stomatal closure [9], and non-stomatal limitations like chlorophyll dysfunction [10], the deprivation of enzymatic proteins and membranes of the photosynthetic apparatus [11], and chloroplast ultrastructure destruction [12], have been reported to have a significant negative impact on plant photosynthesis at high salinization levels; salt-affected soils demonstrate higher Na^+^/K^+^ and Na^+^/Ca^2+^ ratios due to the increased presence of Na^+^ in the soil solution. Consequently, a reduction in the uptake of potassium ions (K^+^) and calcium ions (Ca^2+^) might hinder cellular functioning, resulting in the destabilization of cell membranes and the impairment of enzymatic activity, which is facilitated by the enzymes [13]. The production of too many reactive oxygen substances/species (ROS) in the cytosol, chloroplast, and mitochondria is influenced by osmotic pressure and ionic toxicity, leading to oxidative damage [9,14]. These reactive oxygen species can damage plant tissues, alter the double-helical DNA, disrupt the phospholipid bilayer [15], degrade the active biomolecules like lipids and proteins, and destroy photoreceptors like phytochromes A and B [16] (Figure 1). The crops commonly employed for human and animal sustenance, including cereals such as rice and maize, forages like clover, and horticultural crops such as potatoes and tomatoes, are frequently cultivated using irrigation techniques. However, these crops are susceptible to the detrimental effects of elevated salt concentrations in irrigation water or the soil’s rhizosphere solution. In a significant proportion of farmed plant species, the productivity starts to decline even when exposed to relatively moderate levels of salinity in irrigation water (electrical conductivity of water, ECw > 0.8 dS/m) or soil (electrical conductivity of saturated soil extracts, ECse > 1 dS/m) [3]. The impact of soil salinity on the productivity of several vital crops, including *Zea mays*, *Solanum tuberosum* L., *Lycopersicon esculentum* Mill., and *Oryza sativa* L., has been shown to result in significant reductions in yield. These reductions have been reported as 19.0% for *Zea mays*, 12.0% for *Solanum tuberosum* L., 9.9% for *Lycopersicon esculentum* Mill., and 12% for *Oryza sativa* L. [17].

The upregulation of genes involved in various stress responses, including salt stress, can be controlled by transcription factors (TFs), the terminal transducers in a signaling cascade. TFs accomplish this by locating and attaching themselves to conserved cis-acting areas in the promoter of downstream genes [18,19]. Extensive research over the past few decades has led to the identification and characterization of multiple kinds of TFs. As a result, numerous experiments have been conducted to boost stress tolerance by genetically modifying these TFs. bZIP, AP2/EREBP, MYB, WRKY, NAC (NAM, ATAF, and CUC), and WRKY are among the essential TF families involved in abiotic stress tolerance [20].

Nanotechnology, a new industrial revolution, is an emerging technology undergoing rapid development [21]. Over the past decade, nanotechnology has emerged as a crucial instrument for increasing agricultural productivity. Nanotechnology has the potential to completely transform the agricultural industry in several areas, including soil management, water supply for agriculture, and the detection and treatment of plant diseases [22]. The farm industry is one of developed countries’ most important economic foundations. With an increasing global population comes a growing demand for food and agricultural products. Several factors, including plant diseases, increased environmental pollution, diminishing soil and water supplies, and climate change, make agriculture and producing enough nutrient-dense food difficult [23,24,25,26].

Generally, nanotechnology has the potential to significantly contribute to the industry’s rising prosperity by maximizing the use of agricultural inputs, including water, fertilizer, and pesticides, while lowering effluents and pollution [27]. The total effectiveness of using agricultural inputs, including water, light, and chemicals, can be improved using nanotechnology. By enhancing the microbiome, improving agricultural disease management, and reducing losses, soil and plant health and function improve, causing less environmental collateral damage. As a result, nanotechnology offers hope for the development of sustainable agriculture [28]. Abiotic stressors are environmental conditions that limit a plant’s ability to grow, thrive, and reproduce. Abiotic stressors that naturally affect plants include many types, such as heat, cold, salinity, drought, and heavy metals [29]. Since plants are sessile organisms, they have various coping mechanisms that allow them to adapt to changes in their growing settings and display essential flexibility in responding to environmental challenges without impairing cellular, physiological, or developmental processes [30].

Growing more resilient crops to abiotic stressors is one of sustainable agriculture’s main challenges. ROS accumulate in plant cells to dangerous levels in reaction to abiotic stresses. Excess ROS leads to cellular toxicity, membrane proteins and lipids disintegrating, and a reduction in plant growth. The antioxidant defense system scavenges ROS to lower oxidative stress [31]. Nanoparticles have garnered attention in technological applications due to their distinct properties, such as their diminutive size, expansive surface area, enhanced solubility, and heightened reactivity compared to bulk materials [31]. Metallic nanoparticles (MNPs), including Zn, Fe, Ti, Ag, and Cu-NPs, have garnered significant attention due to their potential application in agriculture without causing adverse environmental effects [32]. They have lately been used to test a variety of plants for stress tolerance, plant development, and germinating seeds [33,34].

This review article aimed to investigate the various adverse effects of salt stress on crop yield, and to gain a deeper understanding of the regulatory and mitigating mechanisms of salt stress in the case of the use of metal oxide nanoparticles—especially zinc oxide nanoparticles—during a stressed state, and to understand the relationship between these artificially produced zinc oxide nanoparticles and transcriptional regulators.

## 2. The Presence of High Salinity Levels Has Been Observed to Result in a Decrease in Biomass Production and Subsequent Yield Losses

When the growth requirements of crop plants are appropriately met, including with effective management practices throughout the growing season, as well as the proper management of nutrients, water, light, and temperature to align with their optimal growth conditions, these plants are capable of yielding high quantities of high-quality grain, fiber, and substances with high sugar, oil, or protein contents. In the context of plant development, it has been observed that abiotic stressors such as salt or drought have an inverse correlation with yield-related characteristics [35]. If salt stress inhibits the growth of plants during the early stages of development, it can significantly reduce the yield and negatively impact the quality and quantity of plant products [36]. While the visible effects of salt may not always be immediately evident, it is crucial to remember that salinity can lead to reductions in agricultural production. A specific crop’s salt tolerance and sensitivity are determined by its ability to extract water and nutrients from saline soils and its capacity to prevent the accumulation of salt ions in its tissues [37,38].

Elevated levels of sodium ions (Na^+^) and chloride ions (Cl^−^) can have detrimental effects on plants, especially when they accumulate within the cytoplasm [39]. Despite the significance of the topic, there remains a limited understanding regarding the specific cytosolic processes that are negatively affected by elevated levels of salt ions, and the potentially harmful effects of chloride in the cytosol remain uncertain [40]. Additionally, the accumulation of deleterious ions in photosynthetically active tissues can accelerate the aging process of transpiring leaves while reducing the availability of essential nutrients required for optimal health [41]. The plant’s capacity to acquire nutrients and energy through photosynthesis or to metabolically utilize them can be limited at any moment [42]. Plants allocate a significant portion of their power to essential processes such as maintenance, vegetative development, and generative growth in the absence of stressors. Nevertheless, with an increasing focus on mitigating stress, resources are reallocated in response to escalating salinity levels.

The energy expenditure is necessary for stress mitigation, growth, and maintenance [43]. The relative proportions of plant constituents vary based on the plant’s developmental stage and susceptibility to salt stress. Plants that are allowed to attain more significant sizes will want increased levels of care and attention. An increase in salinity levels can lead to a decrease in the rate of photosynthesis due to induced stomatal closure, a reduction in gas exchange (CO_2_ input decreased), a reduction in the size and frequency of stomata (after prolonged exposure), the degradation of the chlorophylls, damaging the chloroplast ultrastructure, and the inhibition of photophosphorylation and carbon reactions (enzyme reactions). As a consequence, there will be a reduction in the total energy acquired. To effectively cope with the elevated salt content in the soil, the plant must allocate additional resources toward various processes [2]. These include incurring higher energy investment for ion exclusion or compartmentalization, maintaining ion homeostasis, and detoxifying ROS. These examples are limited in number. Mechanisms related to stress tolerance exemplify this category. Plant growth is impeded by high salinity due to the equilibrium between the losses incurred by the plant and the energy acquired by the plant [44]. The occurrence of tissue senescence is attributed to an imbalance between plant composition and demolishing processes, whereby the latter surpasses the former [45]; this is the reaction of a crop that exhibits both sensitivity and tolerance to elevated soil salt levels [46]. Each type of crop has a different response to saltiness, which can be broken down into three stages [47]: homeostasis, which keeps the rate of growth healthy; eustress, which turns on defense genes; and distress, which slows down metabolism and eventually leads to death. Plant species that exhibit both sensitivity and intolerance may possess the ability to flourish within a limited range of salinity tolerance [48]. In the context of safeguarding vulnerable crops, defensive mechanisms are thought to be triggered at an earlier stage and with a reduced intensity [49], but what occurs is that the defensive ability of tolerant/resistant plant species is more precise because they react faster. They start producing protective compounds more quickly and keep the number of compounds higher. Susceptible plants respond later and produce protective compounds more slowly and for a shorter period of time [50].

## 3. The Negative Effects of Soil Salinity on Crops

Compared to other abiotic stresses that reduce the productivity of agricultural output in dry and semi-arid regions, salinity stress stands out as a challenging obstacle to overcome. This is mainly attributable to the natural conditions in these regions, which promote salinization due to insufficient precipitation to enable the leaching of salts. In other words, these factors cause salinization to occur [26]. In the biphasic model of growth reduction caused by salinization, the bad effects of salt-affected soils are shown by a drop in osmosis and ion cytotoxicity in the first phase. This is followed by the production of ROS and programmed cell death (PCD) in the second phase [51]. These effects are further compounded by the generation of reactive oxygen species (ROS) and disturbances in nutrient equilibrium. This aligns with the theoretical proposition that salt induces a decrease in osmosis during the initial stage and an increase in ion cytotoxicity during the subsequent stage [52].

### 3.1. The Effects of Salinity on the Agricultural Value Measuring Properties of Plant Species

Several elements, including the amount of salt present, the amount of time that has passed, the plant species and variation, the photochemical quenching capability, the plant growth stage, the type of stress, the gas exchange characteristics, and the photosynthetic pigments, all contribute to the retardation of plant development that occurs under salt stress [53]. In several studies on corn (*Zea mays* L.) [54], rice (*Oryza sativa* L.) seedlings [55], *Vigna unguiculata* L. [56], and rapeseed (*Brassica campestris* L.) [57], it was discovered that increasing plant height with even a slight quantity of salinization was beneficial.

The observed increase in plant height can be attributed to fewer soluble salts in the growth control medium; in the beginning, the nutrient salts added to the medium act as eustress on the plant, with a stimulating effect. Conversely, the decrease in plant height can be attributed to the adverse effects of excessive salts on various physiological processes. These effects include a reduction in photosynthetic rate and decreased levels of carbohydrates and growth hormones, which inhibit growth, and decreased protein synthesis due to alterations in antioxidant enzyme activities [58].

Salinity levels of up to 8 dS/m substantially influenced various plant parameters, including fresh and dry weight, leaf area, and leaf count [59,60,61]. Numerous researchers have observed that the reduction in dry matter production and deceleration of plant growth in soils influenced by salinity can be attributed to the hindrance of cell elongation [53] through the direct suppression of the activities of transport proteins like proton (H^+^)-ATPase [62]. Salt stress has been found to have detrimental effects on photosynthesis, leading to constraints in both plant and leaf growth and reductions in chlorophyll levels, as the central core of chlorophyll contains magnesium [63]. The absorption or uptake of magnesium is inhibited by acidic pH. [64]. Furthermore, when subjected to salinity stress, the fresh and dry biomass of *Brassica napus* L. cv. exhibited a significant reduction, with shoot growth being more severely damaged compared to root growth [59]. Based on a conceptual framework, it is hypothesized that salinization causes a reduction in the rate of root growth as well as a restriction in the amount of leaf area expansion that can occur as a result of decreased water uptake efficiency. This process helps maintain an appropriate soil moisture level and prevents an excessive buildup of soluble salts within the soil [59].

The phenomenon of stomatal closure in plants, leading to a decrease in photosynthetic activity and a subsequent reduction in carbon assimilation, could potentially be an additional determinant of the reduced growth rate observed in saline conditions. Based on the results of several investigations, it has been observed that a significant presence of sodium (Na^+^) and chloride (Cl^−^) ions within the cell sap creates a subtle osmotic gradient within the nutrient-rich environment. Consequently, this phenomenon reduces the plant’s water absorption capacity, altering its morphological traits [65]. Previous studies have shown evidence that an increased concentration of salt negatively affects nitrogen accumulation in plants due to the interplay between chloride ions (Cl^−^) and nitrate ions (NO_3_^−^), as well as sodium ions (Na^+^) and ammonium ions (NH_4_^+^). Consequently, this phenomenon results in a decrease in the quantity of plant growth and agricultural productivity [66]. Research has shown evidence that salinity stress has a secondary impact on plant feeding, leading to a notable decline in nutrient absorption due to a fall in osmotic pressure [67].

### 3.2. Physiological Processes Whose Characteristics Are Influenced by Salinity Stress

Recent research indicates that the physiological characteristics of cereal crops, including wheat (*Triticum aestivum* L.) and mung bean (*Vigna radiata* L.), are negatively impacted by soil salinity stress [8,68,69]. Decreasing leaf photosynthetic capacity and concentration of carotenoids and other photosynthetically active pigments, as well as altered energy in the mechanisms of ion exclusion, osmotic adjustment, and nutritional imbalance, may all be contributing factors to the loss in plant growth and output caused by soil salinity [70]. Crops are often impacted by salt-affected soils in one of three ways: oxidative damage, ion imbalance, or osmotic stress [71]. The harmful effects of sodium (Na^+^) and chloride (Cl^−^) ion buildup in plant tissues are the predominant reaction of salt-affected soils [71,72]. It has been demonstrated that plants under salinity stress collect more Na^+^ ions, disturb the ionic balance, modify plant metabolism, and increase oxidative damage. In contrast, the K^+^ ion status in plant tissues aids in the development of soil salt tolerance in plants [73].

When rice (*Oryza sativa* L.) was cultivated in salt-affected soil, the K^+^ ion concentration was only marginally changed; however, the Na^+^ content in the leaves was dramatically increased, and the K^+^/Na^+^ ratio was significantly decreased [71,72]. A notable decline in strawberry plant growth was noted [74]. These growth delays may partly be explained by decreased photosynthetic activity brought on by lower levels of Chl a and Chl b under varying salinity conditions [75]. Ion imbalance in plants and soil is brought on by the entry of Na^+^ and Cl^−^ ions, and this imbalance in the plant’s ions may lead to severe physiological issues [76]. The high salt content in the soil profile may result in physiological dryness as a result of decreased water uptake, salt accumulation in the root zone of the plant, a reduction in plant osmotic potential, and the subsequent disruption of cell metabolic processes as a result of ion toxicity [76,77]. Excessive Na^+^ in plants damages the cell membrane and organelles and impairs physiological processes that result in the death of plant cells [72,78,79]. These physiological processes include the net photosynthetic rate (Pn), stomatal conductance (Gs), transpiration rate (Tr), intracellular carbon dioxide (Ci) concentration, and soil plant analysis development (SPAD) chlorophyll value.

The disturbance of the selective permeability of the cell membrane, which makes it difficult for the plant to detoxify the ROS in the cytoplasm, a decrease in the rate of photosynthetic activity, and alterations in the antioxidant enzymes, may also occur as a result of these physiological changes [78]. These oxidative systems can interfere with dynamic cellular operations in plants under abiotic stress—particularly soil salinity—by disrupting the normal functions of numerous plant cellular components such as proteins, DNA, and lipids [80]. Additionally, plants raised in salty environments may impede chlorophyll production and cause considerable changes to the activities and structure of the pigment–protein complex [81]. The decreased activity of several enzymes, including porphyrinogen IX oxidase, porphobilinogen deaminase, coproporphyrinogen III oxidase, 5-aminolevulinic acid dehydratase, protochlorophyllide oxidoreductase, and Mg-chelatase, may be the cause of the inhibition of chlorophyll pigment synthesis under salt stress [82]. These enzymes, in turn, are in charge of either an increase in chlorophyllase activity [83] or a decrease in leaf water potential, N absorption, and, consequently, the reduced ability of plants to synthesize oxygen [69]. Salinity-induced superoxide radicals and hydrogen peroxide (H_2_O_2_), which deteriorate the membranes of thylakoids and chloroplasts, may also be responsible for chlorophyll degradation [84].

### 3.3. The Variant Effects of Salinity on Enzymatic and Non-Enzymatic Antioxidants in Plants

The effects of soil salinity stress, which inhibits plant growth and development, are accompanied by a significant accumulation of ROS. ROS (O_2_^−^, O_2_, H_2_O_2_, and OH^−^) production under stressful conditions (biotic and abiotic) is a stress indicator at the cellular level and is referred to as a secondary messenger. ROS play their role in the biological activities of plants, ranging from gene expression and translocation to enzymatic chemistry [85,86]. ROS may eventually disrupt the average plant metabolism because of lipids, proteins, and nucleic-acid structure changes [87]. According to some reports, soil salinity-stimulated oxidative stress brought on by the accumulation of increased levels of H_2_O_2_ may cause DNA breakage, chromatin condensation, apoptosis, and cell shrinkage [88]. Higher levels of ROS production during salinity stress may cause the thylakoid membranes to produce more malondialdehyde (MDA). Calculating the lipid peroxidation of plant cells involves using MDA concentration, which is recognized to be a reliable indication of lipid peroxidation [89]. The degree of collateral damage to these molecules involved in plant metabolism is determined by the balance between ROS generation and their removal by the antioxidative defense mechanism [90].

Additionally, soil salinization results in acute oxidative damage to plant tissues. As a result, plants create their own sophisticated natural antioxidant defense mechanism to counteract the oxidative stress caused by salinity. The cell-structural harm brought on by salinity-induced ROS is prevented by antioxidant enzymes [91]; these crop plants that have increasing tolerance to osmotic stress and efficient Na^+^ and Cl^−^ exclusion are thought to be better at withstanding salt than other plant kinds because they have an effective antioxidant system. The varied effects of salinity stress on antioxidative enzymatic and non-enzymatic activities in *Tanacetum parthenium* L. [61], *Brassica napus* L. [59], *Oryza sativa* L. [92], and *Glycine max* L. [93] have previously been documented by many studies. Carotenoids, ascorbic acid (vitamin C), α-tocopherol (vitamin E), flavonoids, and phenolics comprise most of the non-enzymatic antioxidant system. In contrast, peroxidase (POD), superoxide dismutase (SOD), ascorbate peroxidase (APX), glutathione reductase (GR), polyphenol oxidase (PPO), and other enzymes make up the enzymatic antioxidant system. The primary function of the enzymatic antioxidative system is to remove the harmful radicals generated during oxidative stress, aiding agricultural plants in withstanding abiotic stress like salinity [86]. Nearly every component of the plant has some natural antioxidants. These natural antioxidants include vitamins, carotenoids, phenols, dietary glutathione, flavonoids, and endogenous metabolites [94]. Plants’ first line of defense against oxidative stress in soils influenced by salt is the synthesizing and scavenging these antioxidants.

## 4. Different Pathways of Salt Tolerance in Plants

### 4.1. Osmotic Adjustment

Salinity tolerance varies widely among plants, affecting their growth responses. Many salt-tolerant plants adjust mechanisms to counteract water stress (Figure 2) [9]. Plants may need osmotic adjustment to survive excessive salinity [95]. The plant must maintain low cytosolic Na^+^ concentrations, high K^+^/Na^+^ ratios, and cellular turgor at low osmotic potentials to grow in salty soil. These properties support K^+^-dependent metabolic processes through osmotic adjustment [9]. Many appropriate solutes contain nitrogen. These N-containing compounds include glycine betaine, proline, and polyamines [96,97]. Thus, nitrogen metabolism matters under stress.

### 4.2. The Exclusion, Redistribution, or Inclusion/Sequestration of Salt Facing the Ion Toxicity

Some transporters work to prevent Na^+^ from building up and to neutralize its harmful effects in the cytosol. In return for an H^+^, the vacuolar Na^+^/H^+^ antiporter (NHXs) sequesters a sodium ion into the vacuole [98]. Keeping cytosolic Na^+^ levels low at the cellular level and keeping Na^+^ concentrations low at the whole-plant level are efficient ways for glycophytes to deal with salinity stress. Apart from these variables, it was discovered that K^+^ uptake and maintenance significantly affected plants’ ability to withstand salt [99]. It has been strongly suggested that maintaining high cytosolic K^+^/Na^+^ ratios, particularly in shoots, is essential for glycophyte plants to be salt-tolerant [100].

“Sodium exclusion” is frequently used interchangeably with “preventing its accumulation in the shoot”. Na^+^ extrusion from the root thus has an actual role. In animal cells, Na^+^/K^+^-ATPase releases three Na^+^ ions in return for two K^+^ ions [101]. While Na^+^ can be more easily extruded in lower plants by sodium ATPase (PpENA1), in higher plants, the Na^+^/H^+^ antiporter Salt Overly Sensitive 1 (SOS1) is still the only transporter that is known to keep Na^+^ out of the cytosol and into the apoplast [102]. The authors of [103] confirmed that a 10- to 20-fold higher Na^+^ extrusion ability was observed in their study on wheat (under the same considerations) in the root extension zone, with net Na^+^ fluxes ranging from –284 ± 39 nmol m^–2^ s^−1^ to –1584 ± 237 nmol m^−2^ s^−1^, compared with the study by the authors of [104] that reported that mature root zone Na^+^ efflux mostly ranged between –20 nmol m^−2^ s^–1^ and –70 nmol m^–2^ s^–1^.

The mature root zone, which makes up most of the root and can lower the Na^+^ load in the shoot, may provide the physiological explanation for the sequestration phenomena [103]. It has been extensively shown that decreased Na^+^ buildup in the shoot is associated with the high-affinity K^+^ transporter (HKT)-mediated retrieval of Na^+^ from the xylem [105], and last, wheat’s resistance to salt occurs [106]. HKT transporters are constituents of the Trk/Ktr/HKT superfamily and are frequently denoted as monovalent cation transporters. Extensive research has been conducted on these entities across several plant species, revealing their vital role in enhancing salt tolerance. Their primary function involves preventing the infiltration of Na^+^ ions into the vulnerable shoot tissues of plants [107]. There is not much physiological justification for intentionally loading Na^+^ in the shot and retrieving it. By instantly depositing a salt load in the root cortex of the mature zone, this pointless cycle can be broken, and the same result can be obtained at a lesser cost [104]. Notably, wheat’s vacuolar fluorescence Na^+^ signal has a notably higher intensity in the root elongation zone than in the mature zone, implying that Na^+^ may be transferred from the mature root zone to the expanded zone [103].

### 4.3. The Activation of Redox Responses

One of the critical mechanisms by which plants are damaged during adverse environmental conditions is the excess production of reactive oxygen species (ROS) (Figure 3)—the salt-induced ROS that most likely originate from the mitochondrial and chloroplast electron transport chains [98]. Eliminating harmful ROS species is the most crucial component of an effective acclimatization response in salt stress. Reduced stomatal conductance is how plants react to salt stress to prevent excessive water loss. Consequently, this lowers internal CO_2_ concentrations (Ci) and slows down the Calvin cycle’s elimination of CO_2_ [108], causing photorespiration, especially in C3 plants, which causes the peroxisome to produce more H_2_O_2_ [109]. Numerous findings suggest that various chemicals, pigments, and enzymes have defensive functions that reduce oxidative damage and improve plant tolerance to salt by eliminating hazardous ROS. The antioxidant enzymes are SOD, CAT, POX, and APX, while the nonenzymatic antioxidant compounds are glutathione (GSH), ascorbates (ASC), and carotenoids [110]. Some other types of antioxidants are called anthocyanins; they are flavonoids known to accumulate in plants subjected to salt stress [98].

## 5. Exploring Nanoparticle Kinds and Their Ability to Reduce Abiotic Stress

The exceptional characteristics of nanoparticles (NPs), which include their heightened reactivity, enhanced surface activity, and large surface-to-volume ratio, comprise both the nanoparticles’ physical and chemical properties [111]. Typical classifications of nanoparticles (NPs) include many kinds that are typically divided according to concepts like the synthesis method and the compounds or elements used in the synthesis. Many studies on metal and metal oxide-based nanoparticles (NPs) have been conducted in agricultural settings over the past decade [112,113,114]. These research studies have been carried out in several different countries with the goals of increasing crop output and enhancing plants’ flexibility and resilience in the face of abiotic stresses [115,116,117,118]. Researchers have shown a great interest in metal-based nanoparticles and the oxides that correspond to them. These nanomaterials include a wide variety of metals, such as gold, silver, copper, aluminum, and iron; in addition, they form a variety of metal oxides, such as titanium dioxide (TiO_2_; titanium has an oxidation state “OS” = Ti^4+^), cerium oxide (CeO_2_; cerium has an OS = Ce^4+^), iron oxide (FeO; iron has an OS = Fe^2+^), aluminum oxide (Al_2_O_3_; aluminum has an OS = Al^3+^), and zinc oxide (ZnO; zinc has OS = Zn^+2^) [33,34,119].

Plants have significant challenges regarding their capacity for growth and productivity when subjected to abiotic stressors such as heat, cold, humidity, and the toxic effects of heavy metals [30]. Plants exhibit responses to abiotic stressors through several mechanisms, encompassing molecular, morphological, physiological, and biochemical alterations. Scientists have employed the concentration-dependent impacts of magnetic nanoparticles (MNPs) on the growth and development of plants to illustrate their potential in helping plants endure abiotic challenges [120]. Depending on the particular mechanism of action that MNPs possess, they can be applied to plants in various ways, including as a seed-priming agent, soil treatment, or foliar applications. Several studies have shed light on the positive effects that specific ZnO-NPs have on certain plant species when those plants are subjected to various abiotic stresses (Table 1). It has been discovered that zinc oxide nanoparticles (ZnO-NPs) can improve nutrient absorption, regulate the Na^+^/K^+^ ratio, maintain water balance, facilitate ion accumulation, and buffer the adverse effects of abiotic stressors. The elevation of flavonoid, anthocyanin, phenolic, and photosynthetic pigment levels and the overexpression of antioxidant enzymes are all responsible for these benefits. In addition, ZnO-NPs have been shown to reduce the expression of stress markers such as malondialdehyde (MDA) and hydrogen peroxide (H_2_O_2_) while also affecting transcriptional factors.

Foliar Zinc can increase wheat yield in alkaline soils [135]. A study examined how foliar ZnO and its nanoparticle treatments affected safflower growth and yield under various watering regimens [136]. ZnO foliar treatments increased safflower yield. The optimal spraying concentration was found to be 5–10 g L^−1^. The authors discovered that applying ZnO increased crop productivity under various stress scenarios. Photosynthesis and dry matter accumulation increased biomass yield. ZnO foliar treatments increased the number of capitula per plant and the number of seeds.

Nano-fertilizers improve product growth and performance while being environmentally friendly. Thus, molecular strategies using NPs can improve plant nutrition and stress resistance [136]. Nano-fertilizers increase nutrient-use efficiency, reduce chemical fertilizer environmental impact, and minimize fertilization frequency [137]. The study found that safflower morphology and physiology have improved, especially in drought. Addressing Zn shortage, developmental signals like seed oil content increases could explain this production increase [138]. The study revealed that nanomaterials applied by foliar spraying increased growth and yield. This relates to nanomaterial characteristics. After spraying nanoparticles on the plant, they assimilate quickly. Few researchers have examined how nanoparticles affect plant development or stress response, and the behavior of different NPs in plants is unknown [138]. The study discovered that 10 g L^−1^ ZnO-NPs enhanced safflower growth, especially with extended irrigation intervals. This shows that safflower absorbs ZnO-NPs better than their conventional form. Because of their tiny size (1–100 nm), ZnO-NPs are more available to plants than regular ZnO [139].

## 6. Various Nanoparticles and Their Impact on the Genes Responsible for Plant Salt Tolerance: Enzymatic Expression

Nanoparticles produced in a laboratory have the potential to interact with plants in a variety of ways, including chemically and mechanically. The entities’ characteristics, such as their size, surface area, and catalytic qualities, all play a role in the interactions between them. The number of studies that have been conducted on nanoparticles’ effects on molecules is minimal [34,140,141]. Zinc oxide nanoparticles (ZnO-NPs) and other nanoparticles significantly impact numerous plant types as stimulators to the antioxidative defensive mechanisms (Figure 4) and assimilate into the soil around the plant. In contrast, others enter the leaf of the plant and accumulate in the edible sections. Environmentally harmful metal and metal oxide nanoparticles include Zn^2+^, Ag^1+^, Fe^3+^, Al^3+,^ and Ti^4+^ [142]. Silver nanoparticles (Ag-NPs) were found to boost the amounts of antioxidant enzymes in *Brassica juncea*, such as guaiacol peroxidase, catalase, and ascorbate peroxidase, which in turn reduced the activity of reactive oxygen species/substances (ROS) [143]. Following the exposure of *Brassica juncea* to gold nanoparticles (Au-NPs), there was an observed rise in the activity levels of various enzymes, including superoxide dismutase (SOD), catalase (CAT), guaiacol peroxidase, ascorbate peroxidase, and glutathione reductase (GHR) [144].

An observation indicated that plants subjected to treatment with Au-NPs exhibit elevated concentrations of proline and H_2_O_2_. The activity of ascorbate peroxidase, glutathione reductase, and guaiacol peroxidase is enhanced by the presence of gold nanoparticles at concentrations of up to 400 ppm. Specifically, guaiacol peroxidase activity is increased by gold nanoparticles at a concentration of 200 ppm. The molecular responses of arabidopsis plants to silver nanoparticle treatment were investigated using a reverse transcription-polymerase chain reaction (RT-PCR) [145]. A whole-genome cDNA expression microarray was utilized to investigate the transcriptional responses of arabidopsis plants exposed to silver nanoparticles. As a result, a total of 286 genes that exhibited increased expression levels were identified.

Among these genes, certain ones were shown to be associated with oxidative stress and metal responses, such as the vacuolar proton exchanger, superoxide dismutase (SOD), cytochrome P450-dependent oxidase, and peroxidase. In addition to identifying genes associated with plant defense, the study also revealed the presence of around 81 genes that exhibited down-regulation. Some genes involved in regulating auxin, the ethylene signaling system, and systemic acquired resistance (SAR) were identified as being responsible for conferring improved and often enduring protection against various illnesses in the plant kingdom. Identifying proteins that exhibit a reaction to silver nanoparticles has revealed their association with several metabolic processes, such as transcription, protein degradation, the oxidative-stress response system, and the calcium-signaling pathway [146]. After treatment with zinc oxide nanoparticles, 660 up- and 826 down-regulated genes were found in Arabidopsis thaliana. Tomato germination rate and seed germination were enhanced by treatment with multi-walled carbon nanotubes, which increase the expression of genes that respond to stress [147]. The effect of synthesized ZnO-NPs on plant characteristics is presented in Figure 5.

## 7. Transcriptional Factors Affecting Salt Tolerance

Transcriptional factors (TFs) attach themselves to DNA regulatory sequences located in the 5′-upstream region of target genes to regulate the rate at which transcribed genes are stimulated. Consequently, by “turning on” and “off” specific genes that connect with other DNA, TFs play a crucial role. Ultimately, they regulate the manufacture of the transcriptional proteins that alter cellular activity in plant tissues. The highly conserved transcription factors (TFs) aid plants in overcoming salt stress by controlling the expression of specific genes. Gene expression is regulated via transcription factors, affecting cell development. First and foremost, as transcription factors influence alterations in gene transcription, it is critical to comprehend transcription mechanisms. RNA polymerase 2 (RNA PII) is the enzyme that carries out transcription in all eukaryotes. Namely, RNA PII is not able to move by itself. The cis-regulatory region, or the DNA sequence found within the gene, and transcription factors, which are proteins that function as trans-acting factors, regulate the activity of this enzyme. Such DNA segments are known as cis-acting elements, and they control transcription. The six significant TF families associated with stress tolerance are DREB, MYB, NAC, WRKY, and bZIP. Table 2 lists TFs that reduce salinity stress [148,149].

The recognition of the stress signal by membrane-bound receptors in the plant cell is the initial step in activating a signaling cascade in response to abiotic stress (Figure 6). Scientists have recently demonstrated that several proteins localized on the plasma membrane act as putative sensors for abiotic stress signals. These proteins include COLD1 (CHILLING-TOLERANCE DIVERGENCE 1), OSCA1 (reduced hyperosmolality-induced calcium increase1), MSLs (MscS-like proteins), CNGCs (cyclic nucleotide-gated channels), GLR (glutamate receptor-like) channels, histidine Ca^2+^, and reactive oxygen species (ROS), which are examples of second messengers that are transmitted downstream from these sensors after identification. A different set of ROS-modulated protein kinases (PKs) and protein phosphatases (PPs), similar to ROS, are activated by the second messengers [208].

These include MAPK (mitogen-activated protein kinase) cascades, CDPKs (calcium-dependent protein kinases), CBLs (calcineurin-B-like proteins), CIPK (CBL-interacting protein kinase), and numerous other PKs in addition to PPs like some PP2Cs (protein phosphatase 2Cs). The information is then sent downstream by these PKs and PPs, which in turn set off cascades of phosphorylation and dephosphorylation, particularly involving TFs. This ultimately results in the expression of regulatory genes that play a role in the transcriptional regulation of gene expression and signaling cascades, or indirectly in the expression of functional genes involved in cellular protection (Figure 7) [149,208,209,210,211,212,213,214,215,216].

The aforementioned stress-responsive genes are subject to the regulation of their expressions through the activation and deactivation of transcription factors via phosphorylation and dephosphorylation mechanisms. A single transcription factor can regulate a cluster of downstream target genes instead of functional genes. These qualities provide promising candidate genes for the genetic manipulation of complex stress-tolerance traits. The different types of salt-stressed responsive genes and their functions can be summarized in Figure 8.

### 7.1. AP2/EREBP (Apetala2/Ethylene-Responsive Element-Binding Protein) Transcription Factors

Developmental, abiotic, and biotic stress response and other physiological processes are all regulated by the broad family of transcription factors known as AP2/EREBP [217]. The AP2/ERF DNA-binding domain, which is widely conserved, is present in AP2/EREBP transcription factors. This domain is responsible for recognizing and binding to specific DNA sequences known as GCC boxes and DRE/C-repeat elements (CRT) located in the promoter region of their target genes [218]. The existence of a predetermined quantity of AP2/ERF domains has made it possible to divide the AP2/EREBP gene family into four distinct subgroups. The proteins known as DREB (dehydration-responsive element-binding proteins), AP2, RAV (related to ABI3/VP1), and ERF (ethylene-responsive element-binding factors) are all included in this category of proteins [217]. Previous studies have demonstrated that members of the AP2/EREBP gene family, specifically DREB and ERF, play a role in the tolerance of biotic and abiotic stress [151,152,153,154].

### 7.2. WRKY TFs

The presence of the WRKY domain distinguishes WRKY from the other members of the large family of TFs. This domain is around sixty amino acids long, has the WRKY sequence at its amino-terminal, and a putative C_2_H_2_ or C_2_HC zinc finger motif at each carboxyl-terminal end. In the promoter region of their target genes, WRKY transcription factors bind to conserved cis-elements known as W boxes. These W boxes are identified by the sequence (T)(T)TGAC(C/T), and they are located in the W box region [219,220]. The number of WRKY domains present in a WRKY transcription factor (TF) and the particular type of zinc finger-like motif that it possesses can be used to divide this type of transcription factor into one of three separate classes. In Group I, there are two WRKY domains, both exhibiting the C_2_H_2_ motif. In Groups II and III, however, there is only one WRKY domain, which either displays the C_2_H_2_ motif or the C_2_HC zinc finger-like motif. The extensive group II can be subdivided into five subgroups using the peptide sequence as the criterion. These subgroups are designated IIa, IIb, IIc, and IIe, respectively [220]. WRKY genes have been identified to play a role in the biological function of abiotic stress tolerance [221]. Based on a recent study, it was found that JcWRKY, a type II WRKY transcription factor derived from the biofuel plant *Jatropha curcas*, could enhance salt tolerance in transgenic tobacco plants. This enhancement was achieved by facilitating the regulation of reactive oxygen species (ROS) levels through the involvement of salicylic acid (SA) [174].

### 7.3. NAC TFs

The NAC family comprises a sizeable number of members and is unique to higher plants in that only they possess it. The name of the gene that encodes the NAC transcription factor (TF) comes from the combination of the names of three other genes: CUP-SHAPED COTYLEDON 2, arabidopsis transcription activation factor 1/2 (ATAF 1 and ATAF 2), and Petunia hybrid No apical meristem (NAM) (CUC 2). The N-terminal portion of these proteins is characterized by a conserved NAC domain responsible for DNA binding. In contrast, the C-terminal portion of these proteins contains a transcriptional activator domain that can take on various forms and is positioned at the C-terminus. Researchers have uncovered NAC genes in several plant species, including rice, Arabidopsis, foxtail millet (*Setaria italica*), and soybean (*Glycine max*), by a procedure known as whole-genome sequencing [222,223,224]. NAC transcription factors (TFs) have been identified as critical regulators of various biological processes, such as growth, fruit ripening, hormone signaling, and responses to both biotic and abiotic stress [225]. Furthermore, genetic manipulation has been employed to enhance the ability of NAC transcription factors (TFs) to mitigate the impact of abiotic stressors on crops [190].

### 7.4. bZIP TFs

The ability of the bZIP group of transcription factors to bind to DNA is facilitated by the presence of a leucine zipper motif at the C-terminus and a conserved essential region with a basic leucine zipper (bZIP) domain at the N-terminus [226]. These defining properties are located at the C-terminus and N-terminus of the factor, respectively [226]. The bZIP family of proteins has another function, which is to react to abiotic stresses such as high salt concentrations, a lack of water, and cold temperatures, among others. Several studies have shown that abscisic acid (ABA) makes it easier for specific genes to become activated [227]. The findings of the empirical study support these findings. These genes are critically important in regulating the expression of many genes involved in the stress response. Notably, they can accomplish this by identifying and interacting with conserved abscisic acid-responsive elements (ABREs) in the promoter region of these genes [228]. Numerous bZIP TFs have been specified in various plant species; for example, arabidopsis [229], rice [230], corn [231], *Brassica oleracea* [232], and *Solanum lycopersicum* [233]. bZIP TFs can be divided into 11 groups (I–XI) [230]. Members of Group A, also known as ABRE-binding factors (ABFs/AREBs), are thought to have a role in how plants react to environmental stresses such as drought and salinity [18].

## 8. Gene Expression and Zinc Oxide Nanoparticles

Phytohormones influence the responses of many plant species to external stimuli by modulating of the expression levels of several auxin-responsive protein (ARP) genes [211]. Auxin gradients in both spatial and temporal dimensions are frequently modulated by factors such as auxin synthesis, auxin transporters responsible for inflow and efflux, and a diverse array of interrelated hormonal signals [212,213]. In a study conducted by Hezaveh et al. (2019) [132], they applied salt and ZnO-NPs to examine the expression profile of the ARP gene. The presence of salinity induced the generation of ROS, subsequently triggering the activation of auxin oxidases. The expression of several genes associated with growth processes was anticipated to be diminished due to the increased breakdown of auxin facilitated by auxin oxidases.

The synthesis of auxin, the metabolic processes involving nitrogen, and the expansion of root cell tissue are all significantly influenced by the zinc ion, an essential nutrient because it stimulates the biosynthesis of tryptophan [234]. Indole-based auxins are synthesized from tryptophan, so your zinc nanoparticles will indirectly affect elongation growth [235]. This is because it serves as a cofactor for various enzymes, such as oxidases, dehydrogenases, and anhydrases, including peroxidases. This is how it can accomplish this [236]. The cation-exchange capacity of the root is increased via zinc, which makes it easier for the root to absorb essential nutrients, including nitrogen. This, in turn, results in a rise in the total amount of protein. Additionally, zinc is critical to the breakdown of carbohydrates and proteins throughout the metabolic process [237].

According to the findings of a study, the level of Myelocytomatosis Oncogenes (MYC) transcription factors that were noticed to be mainly produced in plants were those that had been exposed to a concentration of 20 mg/L of ZnO-NPs. These transcription factors are involved in various biological processes, including the production of anthocyanins, specific amino acids like tryptophan (Tryp), stomatal differentiation, seed germination, and endosperm degradation. They also play a critical role in activating jasmonic acid-responsive genes, which is a process that is required to produce jasmonic acid-responsive proteins. In environments with high salinity, a lower level of MYC expression was found than in environments with lower salinity. It is possible that the degeneration of the MYC progenitors is to blame for the decrease in their level of focus that occurred after they were exposed to salt [121].

## 9. Conclusions

A sustainable agricultural system takes into account a variety of factors, such as the protection of the natural environment, the improvement of human health, the realization of economic and spiritual benefits for both farmers and consumers, and the capacity to fulfill the requirements for food production posed by an ever-increasing global population. The existence of abiotic stress conditions in the environment presents a significant barrier to the expansion of agricultural production on a worldwide scale. The presence of salt substantially influences the ecological balance of the region and the agricultural output of the vast majority of crops. This is because salinity reduces the amount of water available for plant growth. In addition, it acts as a factor that affects the physicochemical characteristics of the ground. A decrease in agricultural output, insufficient economic gains, and harm to the land are some of the adverse effects of salt on a community. The development of saline impacts results from a complex interaction between morphological, physiological, and biochemical systems, including seed germination, plant growth, and the absorption of water and nutrients. A negative effect on the development of plants occurs when they are subjected to salt stress while still in the early stages of their growth. This finally results in a significant decrease in yield under saline circumstances, which jeopardizes both the quality and quantity of plant products. This highlights how important it is to study the various processes, such as transcriptional control, that govern the numerous physiological responses of plants when confronted with such unfavorable situations.

It has been demonstrated that incorporating particular nanoparticles, such as ZnO-NPs, can reduce the harmful effects of salt stress. It has been shown that these nanoparticles interact with a variety of transcription factors (TFs), which may be triggered or released, and also induce the production of certain hormones and substances that regulate osmotic pressure within plant cells. Consequently, these systems contribute to the maintenance of protein integrity, selective permeability, the rate of photosynthetic reactions, and various other critical physiological activities. Since soils and plants are subject to several abiotic stresses, such as salt, drought, low temperature (chilling or frost strains), and hot temperature, more research is needed to determine how plants might moderate these pressures. Salt stress depletes plant vigor and metabolic function, reducing agricultural yield. It thus emphasizes how important it is to study the multiple factors, including at the transcriptional level, that regulate the many activities of plants in these stressful situations. By working with different transcription factors (TFs) that may be released or activated, or even by inducing specific hormones and compounds that regulate osmotic pressure inside the plant cell, specific nanoparticles, like ZnO-NPs, have shown promising results in reducing salinity stress. This helps to preserve the integrity of the proteins, selective permeability, photosynthetic rate, and other vital processes.

## Figures and Tables

**Figure 1 ijms-25-02654-f001:**
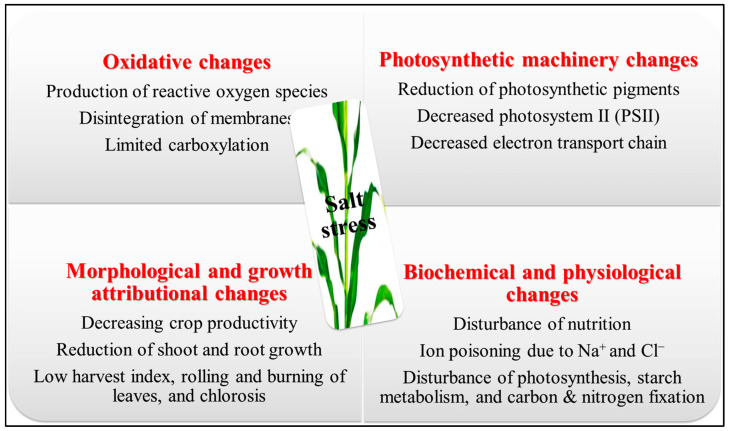
Detrimental effects of salt stress on several plant stages [17]. The image of the maize crop is just an example.

**Figure 2 ijms-25-02654-f002:**
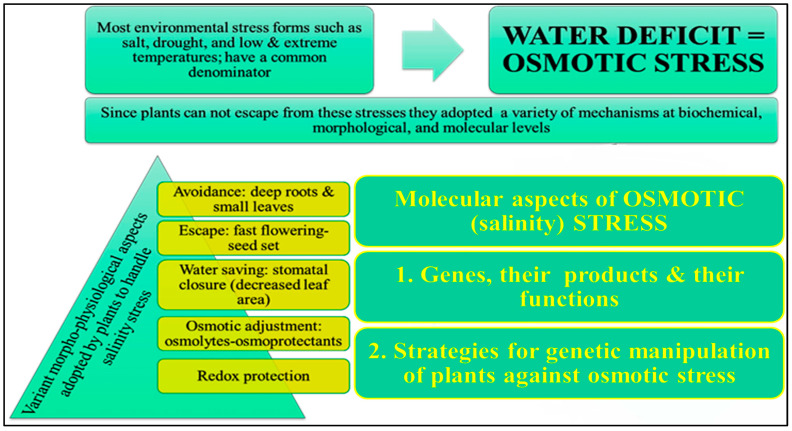
Illustration of different mechanisms that are adapted by the plants against salt stress.

**Figure 3 ijms-25-02654-f003:**
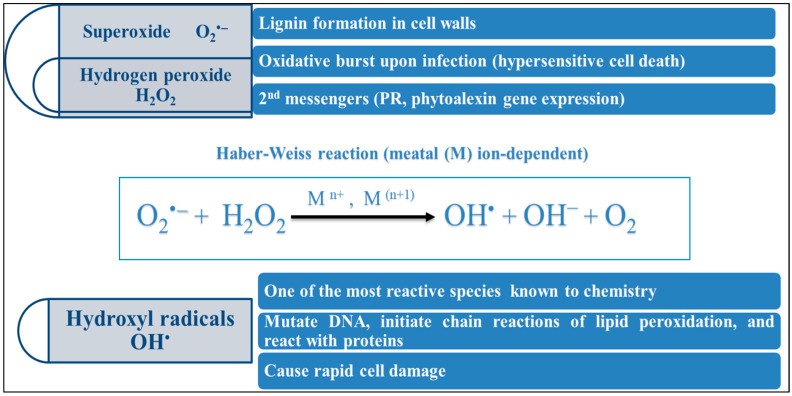
Formation of hazardous ROS under salt-stress conditions.

**Figure 4 ijms-25-02654-f004:**
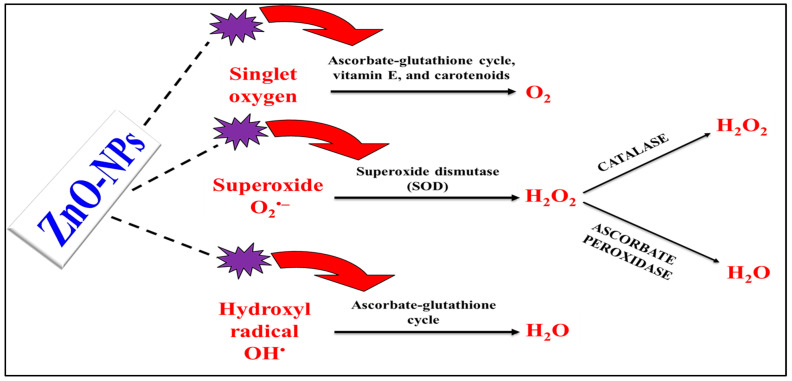
Illustration of proposed know-how of NPs in boosting the antioxidant enzymes to enhance the salt-stressed plants overcoming the harmful effects of ROS. **NPs** = nanoparticles, **OH**^•^ = hydroxyl radical, **H_2_O** = water molecule, **O_2_** = oxygen molecule, and **H_2_O_2_** = hydrogen peroxide.

**Figure 5 ijms-25-02654-f005:**
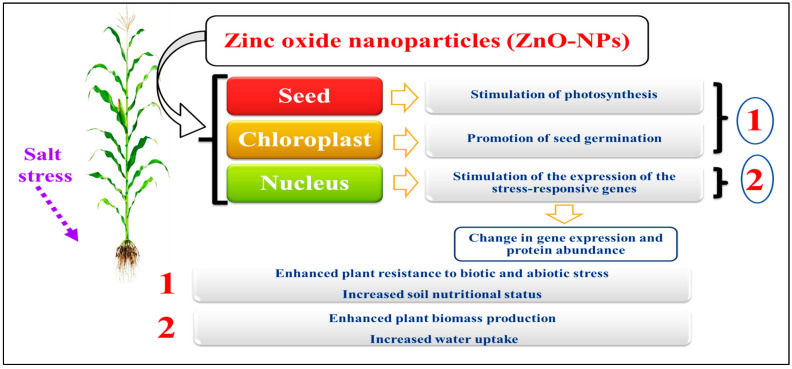
How engineered nanoparticles modify ecophysiological and molecular responses in salt-stressed plants [121]. The image of the maize crop is just an example.

**Figure 6 ijms-25-02654-f006:**
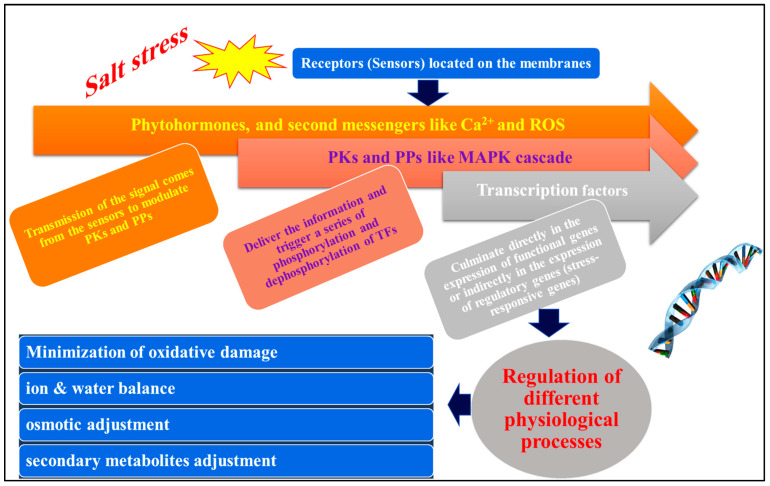
Modeling of signal perception, signal transduction, and gene expression and how the plant responds to salt stress. **ROS** = reactive oxygen species, **PKs** = protein kinases, **PPs** = protein phosphatases, **Ca^2+^** = calcium ions, **MAPK** = mitogen-activated protein kinase.

**Figure 7 ijms-25-02654-f007:**
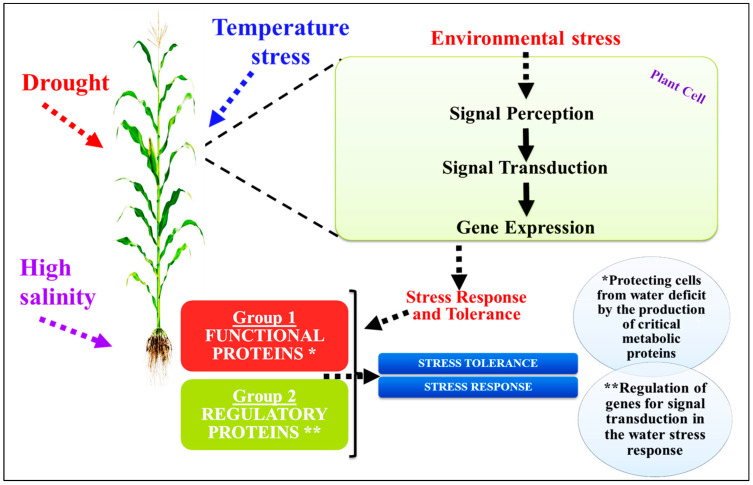
Modeling of signal perception, signal transduction, gene expression, and plant response to salinity stress. The image of the maize crop is just an example.

**Figure 8 ijms-25-02654-f008:**
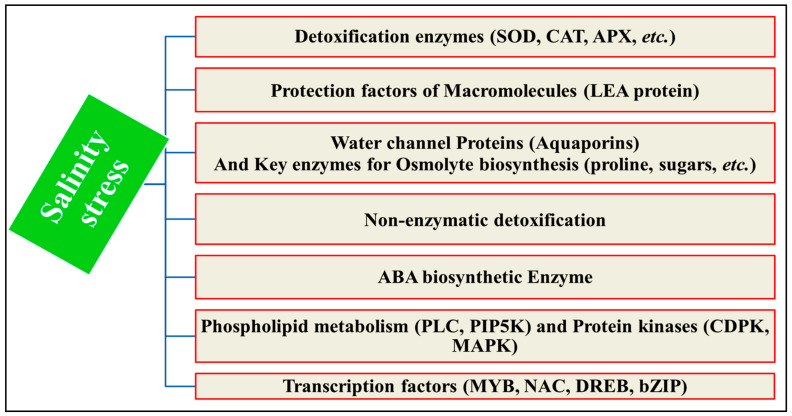
Different types of salt-stress-inducible gene products. **CDPK** = calcium-dependent protein kinase, **CIPK** = calcineurin-B-like interacting protein kinase, **MAPK** = mitogen adenosine protein kinase, **LEA** = late embryogenesis-abundant proteins, **SOD** = superoxide dismutase, **CAT** = catalase, **APX** = ascorbate peroxidase, **POX** = peroxidase, **GPX** = glutathione peroxidase, **GRX** = glutathione reductase, **GOX** = glutathione oxidase, **PLC** = phospholipase C, and **PIP5k** = phosphatidylinositol 4-phosphate 5-kinase, **DREB** = dehydration-responsive element-binding protein, and **ABA** = abscisic acid, **bZIP**, **MYB**, and **NAC** = groups of transcription factors to bind to DNA.

**Table 1 ijms-25-02654-t001:** Investigating the adaptability of certain plant species to salt stress by utilizing zinc oxide nanoparticles (ZnO-NPs) [121].

**Zinc oxide nanoparticles (ZnO-NPs)**	Characters (Size ~nm)	Plant	Mode of Application	Morphophysiological Responses	References
	Spherical and hexagonal (~20 nm), Spherical (80), Spherical (30), or Hexagonal, Square, and Spherical (2–64) [122,123,124,125,126].	*Sorghum bicolor*	Hydroponics, silica sand, foliar fertilization, seed treatment (priming), and foliar spray [122,123,124,125,126].	Reduction in ROS accumulation and lipid peroxidation, improved antioxidant defense system, nutrient absorption, and osmolytes accumulation, seedling development through the biosynthesis of pigments, osmotic protection, reduction of ROS accumulation, adjustment of antioxidant enzymes, and improvement of the nutrient absorption, upregulation of the chilling-induced gene expression of the antioxidant system and chilling-response transcription factors, or reduced MDA content and the elevated level of antioxidant enzyme activities.	[127]
		Tomato (variety PKM-1)			[128]
		Soybean (cv. Giza111)			[129]
		*Glycine max* L.			[130]
		Rapeseed (Okapi cultivar)			[131]
		*Brassica napus* L.			[132]
		*Triticum aestivum* L.			[133]
		*Zea mays*			[134]

**Table 2 ijms-25-02654-t002:** Observed transcriptional genes that exclusively augment salinity stress response in numerous plant species [150].

Family	Genes	Enhanced Tolerance	References
**AP2/EREBP**	*ThDREB, VrDREB2A, SsDREB, TaERF3, EaDREB2, LcERF054, LcDREB2, GmERF7, StDREB1, OsDREB2A, SbMYB15, OsMYB91, SRM1, SbMYB2, SbMYB7, TaMYB3R1, TaMYB19-B, OsMYB48–1, MdSIMYB1, SpMYB, MdoMYB121, OsMYB2, TaMYB73, TaMYB33, and AtMYB15*	Mainly salinity, and they may be involved in drought, NaCl, mannitol, ABA, or cold.	[151,152,153,154,155,156,157,158,159,160,161,162,163,164,165,166,167,168,169,170,171,172,173,174].
**WRKY**	*JcWRKY, MtWRKY76, GhWRKY25, GhWRKY41, GhWRKY68, SpWRKY1, TaWRKY93, TaWRKY44, GhWRKY34, TaWRKY10, ThWRKY4, TaWRKY2, TaWRKY19, OsWRKY08, and OsWRKY45*	Mainly salinity, and they may be involved in drought or cold	[175,176,177,178,179,180,181,182,183,184,185,186].
**NAC**	*VaNAC26, TaNAC47, MusaNAC042, EcNAC67, TaNAC2D, ONAC022, TaNAC29, OsNAC, TaNAC67, SNAC1, GmNAC20, GmNAC11, and ONAC063*	Mainly salinity, and they may be involved in drought or cold	[187,188,189,190,191,192,193,194,195,196,197,198].
**bZIP**	*GhABF2, AtABF3, TabZIP60, OsbZIP71, LrbZIP, ZmbZIP72, ABP9, GmbZIP1, and OsbZIP23*	Mainly salinity, and they may be involved in drought or cold	[199,200,201,202,203,204,205,206,207].

## Data Availability

All data are available within the manuscript.

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
