# Peer review of "The Impact of Salinity on Crop Yields and the Confrontational Behavior of Transcriptional Regulators, Nanoparticles, and Antioxidant Defensive Mechanisms under Stressful Conditions: A Review"

_ijms, 2024, doi:10.3390/ijms25052654_

Round 1

Reviewer 1 Report

Comments and Suggestions for Authors

The review has focused on the impact of salt stress on different aspects of plant growth and on transcriptional regulators.

The abstract should be rewritten including more 'results' or 'conclusion' of the study as it's mainly an introduction except the last sentence

The introduction should be carefully reviewed as the first references provided are not adapted to the sentences associated, as globally they are just too focus on a point not working on the broader topic of the statement. Please see in the attached document where exactly you need to modify the references

Figure 1 has no particular interest as we are not in a physiology book.  No novelty has been provided, neither in figure 3.

Then the rest of the review is well organized and complete, but sometimes some references are missing.

In several figures the acronyms do not appear in the legend

Some figures are deformed and not uniform, they do not like "beautyful".

the conclusions are fine

Comments on the Quality of English Language

Some sentences should be reviewed

Author Response

Responses to Reviewer 1

Thanks so much for your comments. We appreciate your contribution to improving our article. The comments that were mentioned in the PDF document were revised carefully. We took care of every single point and replied to each one inside the PDF document itself, and also modified the needed parts inside the original manuscript. You will find the responses to your respective comments point by point as follows. Thanks again. 

  • The abstract should be rewritten including more 'results' or 'conclusion' of the study as it's mainly an introduction except the last sentence.

The abstract was revised and rewritten so it did not appear as an introduction. More results and conclusions were included.

  • The introduction should be carefully reviewed, as the first references provided are not adapted to the sentences associated, as globally they are just too focus on a point not working on the broader topic of the statement. Please see in the attached document where exactly you need to modify the references.

The introduction was carefully revised according to the instructions in the attached document. The references were also carefully revised.

  • Figure 1 has no particular interest as we are not in a physiology book.  No novelty has been provided, neither in figure 3.

Figures 1 and 3 were removed from the manuscript.

  • Then the rest of the review is well organized and complete, but sometimes some references are missing.

The whole manuscript was carefully revised to ensure the references matched the citations, and other citations were also included as recommended in the attached document. Almost 25 more references were included.

  • In several figures the acronyms do not appear in the legend.

The figures were extensively revised, and the absent acronyms were included in the captions.

  • Some figures are deformed and not uniform, they do not like "beautiful".

Some figures were removed, and others were modified to fit the criteria and look more interesting.

  • the conclusions are fine.

Thanks a lot.

Comments on the Quality of English Language

  • Some sentences should be reviewed.

The spell check, grammar, and structure were revised again, and the manuscript was revised for typos or scientific writing criteria. We hope the revision that was done fulfills the required criteria.

Reviewer 2 Report

Comments and Suggestions for Authors

The present review is quite confuse. Which is the objective?: A general review of salt stress effects in plants? A review on the use of nanoparticles in agriculture? A review of transcription factors related to salt stress? Each topic could be a review by itself, but this review is presented as a patch work difficult to follow, with several redundancies and some topics quite obvius and more adequate for a text book than for a review.

The most interesting and new topic would be the use of nanoparticles in agriculture, so focusing on this, will increase the interest. 

Specific points:

LInes 66-67:  "which is facilitated by specialized organic protein catalysts [12]": this does not make sense. Do you mean "enzymes"?

Lines 67-70: " The production of too many reactive oxygen substances/species (ROS) in the cytosol, chloroplast, and mitochondria is also influenced by other stresses, including oxidative stress followed by osmotic pressure and ionic toxicity [8,13]" This happens exactly the opossite way, osmotic stress and ionic toxicity induce the production of ROS by membrane disruption and impairment of the electron transport chain. 

Line 73: Phytochromes A and B are not pigments.

Figure 1: This figure only shows and open and close stomata. It suggest that this only happens upon salt stress, but this is not the case, as this mechanism is regulated by many physiological processes and diferents kinds of stress. It is so obvious that I would suggest to delete it.

Figure 2: Apparently the review is focused on crop plants, but in the figure you find an Arabidopsis plant. The text in the lower left cornes speaks about crop productivity and harvest. Please, use an image of a real crop. 

line 89: "Production of downregulated genes": Do the authors mean "Upregulation of genes upon stress?".

Line 130: Please, include the oxidation state of the ions when they are found in the nanoparticles. It is incorrect to refer to them as uncharged elements. 

Line 222: rice.

Introduction, and points 1, 2 and 3 are repetitive, and the same information is found in different parts. For instance compare Lines 67-72 with lines 312-317 or lines 155-163 with lines 346-359. Again this general information of salt stress in plants can be found in many reviews, I would suggest to abreviate and consider this as an introduction.

The point 5 is the most interesting part of the present review, although it needs to be improved and updated. For instance: lines 450-451 authors say that "The number of studies that have been done on the effect that nanoparticles have on molecules is extremely limited [31,103,104]", but they have missed some recent articles that fit this description and should be mentioned here such as:

https://www.mdpi.com/2073-4395/13/1/192

Table 1: Fix the tabulation.

Point 7: is quite obvious, would be better as part of the introduction.

Comments on the Quality of English Language

Needs deep revision

Author Response

Responses to Reviewer 2

Thanks so much for your comments. We appreciate your contribution to improving our article. We took care of every single point and replied to each one, and also modified the needed parts inside the original manuscript. You will find the responses to your respective comments point by point as follows. Thanks again. 

  • Lines 66-67:  "which is facilitated by specialized organic protein catalysts [12]": this does not make sense. Do you mean "enzymes"?

Thanks a lot for the comment. Yes, enzymes were meant. We tried to avoid plagiarism in many cases, so sometimes we tried to define some terms. The term "enzymes" was included in the above sentence instead of the definition.

  • Lines 67-70: " The production of too many reactive oxygen substances/species (ROS) in the cytosol, chloroplast, and mitochondria is also influenced by other stresses, including oxidative stress followed by osmotic pressure and ionic toxicity [8,13]" This happens exactly the opossite way, osmotic stress and ionic toxicity induce the production of ROS by membrane disruption and impairment of the electron transport chain. 

We did not mean that at all. Sorry for the misunderstanding; we meant that those kinds of stresses would lead to osmotic pressure, ion toxicity, and oxidative damage, eventually producing reactive oxygen species. The sentence was rewritten to be clearer.

  • Line 73: Phytochromes A and B are not pigments.

Yes, thanks a lot for the comment. Phytochromes A and B are the far-red and red lights photoreceptors mediating many light responses, and despite the phytochrome generally can be defined as a photosensitive pigment that exists in two states, and is involved in photomorphogenic responses, seed germination, bud dormancy, synthesis of gibberellin and ethylene, and photoperiodism, we believe it is better to mention photoreceptors instead of pigments.

  • Figure 1: This figure only shows and open and close stomata. It suggest that this only happens upon salt stress, but this is not the case, as this mechanism is regulated by many physiological processes and different kinds of stress. It is so obvious that I would suggest to delete it.

Thanks a lot. The figure was deleted as recommended.

  • Figure 2: Apparently the review is focused on crop plants, but in the figure you find an Arabidopsis plant. The text in the lower left cornes speaks about crop productivity and harvest. Please, use an image of a real crop. 

Thanks so much for that notice. Another real crop image replaced the image of arabidopsis.

  • line 89: "Production of downregulated genes": Do the authors mean "Upregulation of genes upon stress?".

Thanks so much for the notice. Yes, the upregulation of the stress genes was meant. It is corrected in the manuscript.

  • Line 130: Please, include the oxidation state of the ions when they are found in the nanoparticles. It is incorrect to refer to them as uncharged elements. 

Thanks so much for your comment. Of course, the ions have a totally different oxidation state when they are converted to be included in the new formation nanoparticles. So, the oxidation state of the ions in titanium dioxide (TiO2), cerium oxide (CeO2), iron oxide (FeO), aluminum oxide (Al2O3), and zinc oxide (ZnO) were included in the manuscript.

  • Line 222: rice.

The word was corrected.

  • Introduction, and points 1, 2 and 3 are repetitive, and the same information is found in different parts. For instance compare Lines 67-72 with lines 312-317 or lines 155-163 with lines 346-359. Again this general information of salt stress in plants can be found in many reviews, I would suggest to abreviate and consider this as an introduction.

Thanks so much for your notice. It is highly appreciated. We just want to confirm that we aimed to clarify every title and subtitle with a detailed explanation from our point of view. For that, it sometimes seems that some information was repeated, but that was not the point; the point was to explain every single point in a full perspective. For example, we had to indicate the process of osmotic adjustment under the subtitle ‘different pathways of salt tolerance in plants’, or by demonstrating the effect of ROS on bio-compounds under the title ‘Variant effects of salinity on enzymatic and non-enzymatic antioxidants in plants’, so you may find some repeated meanings because of that need. We appreciate the recommendation so much, and the paragraphs mentioned above were shortened and merged, and the meaning was just kept in the introduction.

  • The point 5 is the most interesting part of the present review, although it needs to be improved and updated. For instance: lines 450-451 authors say that "The number of studies that have been done on the effect that nanoparticles have on molecules is extremely limited [31,103,104]", but they have missed some recent articles that fit this description and should be mentioned here such as:

https://www.mdpi.com/2073-4395/13/1/192

Thanks so much for the kind suggestion. The reference mentioned above was very useful. It was used to improve that part under the title ‘exploring nanoparticle kinds and their ability to reduce abiotic stress’. That section was carefully revised and improved.

  • Table 1: Fix the tabulation.

It is really hard to include the whole needed data in a table that fits the format of the journal. But your suggestion is highly appreciated, and we tried our best to fix it.

  • Point 7: is quite obvious, would be better as part of the introduction.

Your suggestion is really appreciated. But if we shifted the whole seventh section of the article to be included in the introduction, it would be so long, and the backbone of the article would be affected, as we assumed from the beginning to summarize the types, roles, and secretions of the transcriptional factors, then to classify each type with uses and productions in salt-stressed plants. So, your comment is highly appreciated, but we prefer to have that section. We may shorten the content covered by that point if it seems better for you. As we previously reported, we aimed to explain some points in some detail from our point of view, so sometimes you can see that there are some long phrases or repeated meanings, but as we also reported, the repeated meanings were carefully checked to make the manuscript matching with the suggestions of the reviewers.

Comments on the Quality of English Language

  • Needs deep revision.

The spell check, grammar, and structure were revised again, and the manuscript was revised for typos or scientific writing criteria. We hope the revision that was done fulfills the required criteria.

Round 2

Reviewer 1 Report

Comments and Suggestions for Authors

Authors have provided version of the review with significant changes provided increasing the quality of the document particularly on the abstract and the first third of the document where it was needed.

There are still some references to be changed, as there are still not enough adapted to the text, and some english errors to be solved

You can find were exactly in the attached document

I've appreciated the suppresion of some of the figures, but the others could be in terms of form (not content) ameliorated to give a more proffesional aspect, with no distorsion of the letters for example

Comments on the Quality of English Language

A careful read should be conducted to check any other error

Author Response

Responses to Reviewer 1 (Round 2)

Thanks so much for your comments. We appreciate your contribution to improving our article for the second round.

  • Authors have provided version of the review with significant changes provided increasing the quality of the document particularly on the abstract and the first third of the document where it was needed.

Thanks so much for your comment. Your suggestions were very useful and highly appreciated.

  • There are still some references to be changed, as there are still not enough adapted to the text, and some english errors to be solved. You can find were exactly in the attached document.

Thanks a lot for your comment. Your suggestions to improve our article will be always appreciated. Firstly, we appreciate your notice so much, especially for the references and citations and for the other suggestions related to the abstract, figures, and some other points that were very important for us to modify and improve. Secondly, we want to clarify that the attached PDF document already contains the same points of references and citations that were mentioned in the first PDF document you attached for the first round. Those mentioned issues regarding the absent citations or the references that did not fit the mentioned sentences were actually modified and changed in the revised manuscript that was re-submitted. We took care of every single point you suggested to modify, add, or remove. Kindly take a look at the re-submitted manuscript Word document on the system; it will be highly appreciated. You will notice that we modified many citations, added almost 25 new references, and removed some unmatched citations. However, we guaranteed that we would revise the citations and the references one more time to make sure that all references are relevant to the contents of the manuscript.

I am attaching some screenshots from the attached PDF document, followed by the modified sentences included in the resubmitted manuscript. By comparison, you will find that those mentioned points were actually updated as you recommended and suggested. (You will find the screenshots in an attached PDF document).

We also want to declare that all references and citations were written according to the MDPI system, and we used the software “Zotero” to include all the references in the manuscript.

  • I've appreciated the suppresion of some of the figures, but the others could be in terms of form (not content) ameliorated to give a more proffesional aspect, with no distorsion of the letters for example.

Thanks a lot for your comment. By the first round, we changed/modified some figures, and some others were just rearranged. Now, we declare that all the figures were modified to give a more professional aspect and fit the criteria. We are grateful for your endeavor effort in improving our article. Thanks again.

Comments on the Quality of English Language

A careful read should be conducted to check any other error.

Thanks again for your respective notice. The manuscript was carefully read again, and the English language was double-checked on the level of typos, grammar, and structure.

Reviewer 2 Report

Comments and Suggestions for Authors

Paper has been substantially improved. I can recommend publication.

Comments on the Quality of English Language

English has been improved.

Author Response

Responses to Reviewer 2 (Round 2)

  • Paper has been substantially improved. I can recommend publication.

Thanks so much for reviewing our article and for your suggestion and recommendation of publishing. It was our pleasure. Your comments were very useful in improving our article. Thanks again.

Comments on the Quality of English Language

  • English has been improved.

Thanks so much for your kind comment.
